# Hydroxytyrosol Decreases LPS- and α-Synuclein-Induced Microglial Activation In Vitro

**DOI:** 10.3390/antiox9010036

**Published:** 2019-12-31

**Authors:** Marta Gallardo-Fernández, Ruth Hornedo-Ortega, Isabel M. Alonso-Bellido, José A. Rodríguez-Gómez, Ana M. Troncoso, M. Carmen García-Parrilla, José L. Venero, Ana M. Espinosa-Oliva, Rocío M. de Pablos

**Affiliations:** 1Departamento de Nutrición y Bromatología, Toxicología y Medicina Legal, Área de Nutrición y Bromatología, Facultad de Farmacia, Universidad de Sevilla, C/Profesor García González, 2, 41012 Sevilla, Spain; mgfernandez@us.es (M.G.-F.); amtroncoso@us.es (A.M.T.); mcparrilla@us.es (M.C.G.-P.); 2MIB, Unité de RechercheŒnologie, EA4577, USC 1366 INRA, ISVV, Université de Bordeaux, 33882 Bordeaux, France; ruth.hornedo-ortega@u-bordeaux.fr; 3Instituto de Biomedicina de Sevilla (IBiS), Hospital Universitario Virgen del Rocío/CSIC/Universidad de Sevilla, 41013 Sevilla, Spain; isaalobel@gmail.com (I.M.A.-B.); rodriguez@us.es (J.A.R.-G.); jlvenero@us.es (J.L.V.); depablos@us.es (R.M.d.P.); 4Departamento de Bioquímica y Biología Molecular, Facultad de Farmacia, Universidad de Sevilla, 41012 Sevilla, Spain; 5Departamento de Fisiología Médica y Biofísica, Facultad de Medicina. Universidad de Sevilla, 41013 Sevilla, Spain

**Keywords:** hydroxytyrosol, microglia, Mediterranean diet, inflammasome, MAPKs, lipopolysaccharide, α-synuclein

## Abstract

Neuroinflammation is a common feature shared by neurodegenerative disorders, such as Parkinson’s disease (PD), and seems to play a key role in their development and progression. Microglia cells, the principal orchestrators of neuroinflammation, can be polarized in different phenotypes, which means they are able to have anti-inflammatory, pro-inflammatory, or neurodegenerative effects. Increasing evidence supports that the traditional Mediterranean dietary pattern is related to the reduction of cognitive decline in neurodegenerative diseases. A considerable intake of plant foods, fish, and extra virgin olive oil (EVOO), as well as a moderate consumption of red wine, all characteristic of the Mediterranean diet (MD), are behind these effects. These foods are especially rich in polyphenols, being the most relevant in the MD hydroxytyrosol (HT) and their derivatives present in EVOO, which have demonstrated a wide array of biological activities. Here, we demonstrate that HT is able to reduce the inflammation induced by two different stimuli: lipopolysaccharide and α-synuclein. We also study the possible molecular mechanisms involved in the anti-inflammatory effect of HT, including the study of nuclear factor kappa B (NF-κB), mitogen-activated protein kinases (MAPKs), nicotinamide adenine dinucleotide phosphate (NADPH) oxidase, and inflammasome. Our data support the use of HT to prevent the inflammation associated with PD and shed light into the relationship between MD and this neurological disorder.

## 1. Introduction

Parkinson’s disease (PD) is the second most prevalent neurodegenerative disease, following Alzheimer’s disease (AD). Two histopathological features are present in PD: Lewy bodies and Lewy neurites, both formed by inclusions of α-synuclein (α-syn) that are closely related with neuronal death of the nigrostriatal dopaminergic system [1]. There is also strong evidence supporting that neuroinflammation is a pathological condition present in the central nervous system (CNS) leading to neuronal cell death of patients suffering from PD and AD [2]. The most compelling evidence supporting this view comes from multiple genome-wide association studies (GWAS), which have found a genetic association with PD risk including the human leukocyte antigen (HLA) region, tumor necrosis factor (TNF), TNFR1, and interleukin (IL)-1β, among others [3]. 

Microglial cell population plays a key role in brain immune response, defending, and maintaining homeostasis against diverse pathological situations. Brain neuroinflammation is orchestrated by microglial cells, the resident phagocytes that act as the initial responders to pathogens or tissue damage. In response to different environmental cues, microglial cells are activated and their homeostatic molecular state and functions are disturbed. Toll-like receptors (TLRs), one of the main drivers of microglia activation, trigger several transduction pathways, such as the nuclear factor kappa B (NF-κB) pathway and mitogen-activated protein kinases (MAPKs) pathway, among others [4], that cause increased expression of inflammatory cytokines [5]. Moreover, microglia can be polarized into different phenotypes: the classically named M1 (or pro-inflammatory) and M2 (or anti-inflammatory) phenotypes, depending of the stimulating agent. In addition, both phenotypes can coexist or switch from one to another during the inflammation process [6]. However, recent massive transcriptome studies have demonstrated the existence of different microglia subtypes under diverse disease conditions, challenging the concept of M1/M2 microglia polarization states. 

The pro-inflammatory microglia state is defined by the release of cytokines such as IL-1β, IL-6, IL-18, and TNF-α, as well as reactive oxygen species (ROS), reactive nitrogen species (RNS), and nitric oxide (NO), by activating the enzymes nicotinamide adenine dinucleotide phosphate oxidase (NADPH oxidase) or inducible nitric oxide synthase (iNOS) [7]. However, anti-inflammatory microglia are involved in neurodegeneration and tissue repair by suppressing IL-12 secretion and by inducing the release of IL-10, transforming growth factor (TGF)-β, and arginase-1, instead of iNOS [8]. More recently, a new microglia phenotype, called disease associated microglia (DAM), has been related to neurodegenerative conditions [9,10]. In view of the foregoing, suppression of microglia-mediated inflammation can be considered as an important strategy in neurodegenerative disease prevention.

In this context, diet interventions and specifically the adherence to Mediterranean diet (MD) have been demonstrated to be a favorable condition, highly associated with the prevention of neurodegenerative disorders [11,12]. This can be explained by the high proportion in this diet of fruits, vegetables, and olive oil, sources of several polyphenolic bioactive compounds [13]. Hydroxytyrosol (HT) is one of the major compounds present in the phenolic fraction of virgin olive oil, together with tyrosol and their secoiridoids derivatives [14]. Additionally, HT is a product derived from the secoiroids (i.e., oleuropein) by progressive hydrolysis and metabolic transformation after absorption in the gastrointestinal tract. Hence, more relevant plasma concentrations of HT in comparison with other olive oil polyphenols can be expected [15].

A great number of studies have shown that HT exerts a wide array of biological activities such as antioxidant, anti-inflammatory, antithrombotic, antitumor, antiatherogenic, antidiabetic, antiobesity, and antimicrobial agent [15,16].

The anti-inflammatory activity of HT has been previously tested on macrophage cell lines, with the TLR-4 ligand lipopolysaccharide (LPS) as a stimulating agent, and proved that HT diminished the secretion of cytokines induced by LPS, including IL-1α, IL-1β, IL-6, IL-12, and TNF-α, as well as chemokines, such as C-X-C motif chemokine 10 (CXCL10/IP-10) and monocyte chemoattractant protein 1 (MCP-1/CCL2), through a mechanism non-dependent on the NF-κB signaling pathway [17,18,19]. Another recent study has proved that HT in LPS-treated RAW264.7 cells modulates oxidative stress by the involvement of nuclear factor erythroid 2-related factor 2 (Nrf2) [20]. However, to our knowledge, the effect of HT has never been tested in microglia, the leading actors driving the brain immune response under disease conditions.

Taking into account this background and the ability of HT to pass through the blood brain barrier [21], the aim of the present work was to investigate the anti-inflammatory effects in the brain of HT at concentrations compatible with a traditional MD.

Circulating hydroxytyrosol depends on the dietetic hydroxytyrosol intake and endogenous formation. To estimate the dietetic intake, it has to be taken into consideration that major sources are extra virgin olive oil, olives, and wine. If they are consumed within a balanced Mediterranean diet and with the reported values of these foods in mind, the dietary contribution can be estimated as follows: Between 5–8.9 mg of HT can be supplied with a daily consumption of 20 mg of virgin extra olive oil [22,23]. If we consider a normal daily consumption of 40 g of virgin extra olive oil, 2 glasses of wine (6.2 mg/HT) [24] and olives (20 g) (11 mg/HT) [25,26], an amount of 17.25 mg/HT can be provided by the diet. Calculations also considers the bioavailability data (44% approximately) and a dilution in 5 L of plasma. Therefore, the circulating HT from the diet can be up to 10.69 micromolar.

Furthermore, ethanol intake can increase circulating HT as it interferes with the dopaminergic system. If HT is administered with ethanol, dopamine metabolism produces HT instead of DOPAC (3,4-dihydroxyphenylacetic acid), and thus the urinary excretion of HT is at a maximum after red wine intake [27]. In fact, it is quite difficult to distinguish how much HT comes from exogenous or endogenous intake. All in all, because enogenous HT is unknown, the values used in this paper (from 0 to 50 micromolar in plasma) are not far from the values that circulating HT might reach. 

The cell line used was BV2 cells, a microglial cell line that recapitulates many of the original features of primary microglia [28], thus it is the most frequently used substitute for primary microglia. This cell line is especially useful to test the protective role of different compounds on microglial activation [29]. Using both LPS and aggregated α-syn as stimulating agents, we examined the possible molecular mechanisms involved in the anti-inflammatory effect of HT, including the study of NF-κB, MAPK, NADPH oxidase pathways, and inflammasome. Our results show that HT is able to prevent most of the immune-associated alterations induced by LPS and aggregated α-syn through different mechanisms.

## 2. Materials and Methods 

### 2.1. Cell Culture

For all experiments, we used murine microglial BV2 cell line, a gift from Prof. B. Joseph (Karolinska Institutet, Sweden). These cells were cultured in dulbecco modified Eagles minimal essential medium(DMEM) (Invitrogen, Carlsbad, CA, USA) supplemented with 10% heat-inactivated fetal bovine serum (Sigma-Aldrich, San Luis, MO, USA), streptomycin (100 mg/mL), and penicillin (100 IU/mL), under 100% humidity and 5% CO_2_. Experiments were performed in reduced 5% heat-inactivated fetal bovine serum media. All experiments were carried out in 3–20 passage cells.

For testing the effect of HT, BV2 cells were treated with LPS (pro-inflammatory phenotype inducer, 1 µg/mL, Sigma-Aldrich, San Luis, MO, USA; dissolved in phosphate buffer saline (PBS)) or α-syn fibrils, human, recombinant (5 μM, dissolved in PBS; Alexotech, Umeå, Sweden), with and without HT (1, 10, 25, and 50 μM, dissolved in dimethyl sulfoxide, Sigma Aldrich, Steinheim, Germany) for 6 h—the time point where expression levels of cytokines peak after treatment with LPS (PMID: 15772982).

### 2.2. Cytotoxicity Assay

Neutral red (NR) uptake was performed in the BV2 cells. BV2 cells were seeded at a concentration of 12 × 10^3^ cells/well in 96-well plates (Sarstedt AG & Co, Nümbrecht, Germany) for 24 h at 37 °C, and then treated with different concentrations of HT (1, 10, 25, and 50 μM). Treatments were conducted for 6 h. Then, the culture medium was replaced by 100 µL modified medium without serum containing 10 mg/mL NR. The plate with the NR-containing medium was returned to the incubator for another 3 h to allow the uptake of NR into the lysosomes of viable cells. Thereafter, medium was removed, and cells were fixed for 2 min with a formaldehyde-CaCl_2_ solution. By adding 200 µL of acetic acid ethanol solution to the wells, NR absorbed by cells was extracted, solubilized, and quantified at 540 nm.

### 2.3. Real-Time RT-PCR

BV2 cells were seeded at a concentration of 1 × 10^5^ cells/well in 24-well plates (Sarstedt AG & Co., Nümbrecht, Germany) for 24 h at 37 °C, and then treated with LPS (1 µg/mL) or α-syn fibrils (5 μM), with and without HT (1, 10, 25, and 50 μM). Treatments were conducted for 6 h. For the inflammasome activation, cells were treated with LPS or α-syn fibrils, with or without HT (50 µM), for 3 h, and then with ATP (1 mM) for 1 h as the second signal for inflammasome activation [30]. At the end of the treatment, cells were collected in TRIsure™ (Bioline, USA Inc., Swedesboro, NJ, USA) and RNA was extracted from cells following the manufacturer’s protocol. cDNA was synthesized from 1 μg of extracted RNA using Revert Aid First Strand cDNA Synthesis Kit (Thermo Fisher Scientific, Waltham, MA, USA) in 20 μL reaction volume, as described by the manufacturer.

Real-time PCR was performed using 5 µL SensiFASTTM SYBR NO-ROX KIT (Bioline, Swedesboro, NJ, USA), 0.4 µL of each primer, and 4.2 µL cDNA to get to a final reaction volume of 10 µL for the 384-well plate. Controls were carried out without cDNA. Amplification was run in a Lightcycler^®^ 480 Instrument ΙΙ (Roche, Rotkreuz, Switzerland.) thermal cycler at 95 °C for 2 min, followed by 40 cycles consisting of a denaturation phase for 5 s at 95 °C, a second phase of hybridization at 65 °C for 10 s, and a final phase of elongation at 72 °C for 20 s. The process was terminated by a final step of 7 min at 72 °C. Analysis confirmed a single PCR product. β-actin served as reference gene and was used for samples normalization. The cycle at which each sample crossed a fluorescence threshold (Ct value) was determined, and the triplicate values for each cDNA were averaged. The primer sequences for TNF-α, iNOS, IL-1β, IL-6, arginase, CXCL10, the NAPDH subunits (p22^phox^, p47^phox^, and gp91^phox^), nod-like receptor family pyrin domain containing 3 (NLRP3), and β-actin are shown in Table 1.

### 2.4. Western Blot

In order to obtain proteins from BV2 cell line, cells were seeded in six-well plates (Sarstedt AG & Co., Nümbrecht, Germany) at a concentration of 150 × 10^3^ cell/well and left for 24 h at 37 °C to reach the required confluency. Then, cells were treated with LPS (1 µg/mL) or α-syn fibrils (5 μM), with and without HT (50 μM), for 6 h. For the inflammasome activation, cells were treated with LPS or α-syn for 3 h, and then with ATP (1 mM) for 1 h as the second signal for inflammasome activation [30]. At the end of the treatment, cells were collected in 50 μL RIPA buffer (Sigma-Aldrich, St Louis, MO, USA) with the aid of a scraper after being washed three times with ice-cold PBS. The protein lysates were mixed with NuPAGE lithium dodecyl sulfate sample buffer, NuPAGE dithiothreitol (DTT) (Invitrogen, Loughborough, UK), and denatured by heating at 70 °C for 10 min. The protein contents were then subjected to electrophoresis on NuPAGE 4–12% Bis-Tris gels (Invitrogen, Waltham, MA, USA) before being transferred to 0.2 µm nitrocellulose membranes (Bio-Rad, Hercules, CA, USA). Membranes were blocked with 5% bovine serum albumin (BSA) in Tris-buffered saline with Tween 20 (TBST) buffer and incubated overnight at 4 °C with the following antibodies: anti-p38, anti-P-p38, anti-c-Jun N-terminal kinase (JNK) 1/2, anti-P-JNK 1/2(Santa Cruz Biotechnology, Dallas, TE, USA), anti-IL-1β (R&D Systems, Minneapolis, MI, USA), anti-AKT, anti-P-AKT (Santa Cruz Biotechnology), and the housekeeping anti-glyceraldehyde-3-phosphate dehydrogenase (GAPDH) (Santa Cruz Biotechnology). The membranes were then incubated for 1 h at room temperature with anti-rabbit antibodies (Cell signaling Technology, Danvers, MA, USA) in 5% BSA in TBST. The immunoreactive bands were detected using SuperSignalWest Pico chemiluminescent substrate (Thermo Scientific, Hitchin, UK) and visualized on an Amersham Imager 600 station (GE Healthcare live sciences, Marlborough, MA, USA).

### 2.5. ROS Production 

To determine the level of oxidative stress produced in viable BV2 cell cultures, 35 × 10^3^ cells/well were seeded in Ibidi^®^ plates (Invitrogen, Carlsbad, CA, USA) and incubated for 24 h at 37 °C. Then, cells were treated with LPS (1 μg/mL) or α-syn fibrils (5 μM), both with and without HT (50 μM), for 6 h. At the end of the treatment, a membrane permeable reagent was added (CellROX^®^ Deep Red Reagent (Thermo Fisher Scientific, Waltham, MA, USA) to each well. This reagent is a fluorogenic probe that exhibits fluorescence in the presence of ROS in the cell cytoplasm.

The test was performed following the manufacturer’s instructions. The ROS sensitive reagent was added to the treated cells to reach a final concentration of 5 mM, and incubated at 37 °C for 30 min. Subsequently, the media was removed and cells were washed with phosphate buffer saline (PBS) three times. Confocal microscope (Zeiss LSM 7 DUO, Munich, Germany) was employed to acquire images of the treated cells. Images of five fields per condition were captured. These images were later analyzed using the ImageJ software. In each image, the total number of cells was calculated, and the area integrated intensity for fluorescent and non-fluorescent cells was measured. Then, the following formula was used to compute the corrected total cell fluorescence (CTCF):CTCF = Integrated density − (Area of selected cell × Mean fluorescence of background readings).

### 2.6. Immunohistochemical Detection of NFκB p65 Nuclear Translocation

To determine the level of NF-κB p65 nuclear translocation in viable BV2 cells, 40 × 10^3^ cells/well were seeded in 24 well-plates (Invitrogen, Carlsbad, CA, USA) and incubated for 24 h at 37 °C. Then, cells were treated with LPS (1 μg/mL) or α-syn fibrils (5 μM), both with and without HT (50 µM), for 6 h. Paraformaldehyde-fixed cells were blocked with PBS/5% bovine serum albumin/0.1% triton X-100 and incubated in the blocking solution with rabbit polyclonal NF-kB p65 antibody (4 °C, overnight; Cell Signalling Technology, Danvers, MA, USA). After washing, cells were incubated with AlexaFluor 488 conjugated anti-IgG used as secondary antibody (room temperature, 1 h; Molecular Probes). Nuclei were counterstained with Hoechst 33342 (1 µg mL^−1^, Molecular Probes) and preparation was mounted with fluorescence mounting medium (Dako, Glostrup, Denmark). As a negative control, procedure cells were also stained with blocking solution without primary antibody. The results were analyzed using a Zeiss LSM 7 DUO confocal laser scanning microscope (Carl Zeiss MicroImaging Gmbh, Munich, Germany). The nuclear translocation of NF-kB p65 was quantified using ImageJ software (National Institutes of Health, Bethesda, MA, USA) and measured as the fold change of expression level of NF-kB p65 in the nucleus of treated versus control cells. Background level of secondary antibody staining was previously subtracted to all samples under study.

### 2.7. Statistical Analysis

The results were expressed as mean ± SEM. Means were compared by one-way analysis of variance (ANOVA) followed by the LSD test for post hoc multiple range comparisons. An alpha level of 0.05 was used. The Statgraphics Plus 3.0 statistical package was used for the analyses.

## 3. Results

### 3.1. Cytotoxicity of HT

The cytotoxicity of HT at different concentrations (1, 10, 25, and 50 µM) was carried out to select the non-cytotoxic concentration to use in the following experiments. Viability was the primary metric that was used to test cell cytotoxicity. As can be observed in Figure 1, HT was not toxic for BV2 cells at any of the tested concentrations.

### 3.2. HT Reduces Microglial Activation In Vitro

As none of the doses of HT tested had cytotoxic effects, we proceeded to test which of them had a greater effect in reducing the microglial activation induced by LPS and α-syn. The selected dose of HT used for further experiments aimed to discern the mechanism of action of HT. For this purpose, RT-PCR analyses of pro-inflammatory (TNF-α, iNOS, IL-1β, IL-6, and CXCL10) and anti-inflammatory (arginase) mediators were performed.

After treatment with LPS, a strong induction of the five pro-inflammatory markers studied was found, ranging from 3.29-fold change for iNOS to 65.63-fold change for IL-6 (with respect to control levels; *p* < 0.05; Figure 2A–E). However, treatment with HT was able to reduce the expression of all of them (*p* < 0.05; vs. LPS levels; Figure 2A–E). The ANOVA indicates a significant effect of the dose in IL-1β, IL-6, and CXCL10 mRNA expression levels. With regard to Arginase, no effect was observed neither with LPS nor with any dose of HT (Figure 2F).

In the same way, we verified the effect of HT on the induction of inflammatory mediators by α-syn. As can be observed in Figure 3, treatment with α-syn produced a strong expression of most pro-inflammatory mediators, ranging from 9.91-fold change for TNF-α to 19.05-fold change for IL-6 (vs. control levels; *p* < 0.01; Figure 3A–E). Again, HT treatment was able to reduce the expression levels of pro-inflammatory mediators induced by α-syn (*p* < 0.01; Figure 3A–E). Arginase levels, however, remained unaltered in response to α-syn/HT treatment (Figure 3F).

### 3.3. Effect of HT on the Induction of NADPH Oxidase and ROS Production

NADPH oxidase is an enzymatic complex consisting of several subunits, including cytosolic subunits (p40^phox^, p47^phox^, and p67^phox^), the membrane bound cytochrome b558 (p22^phox^), the heme binding enzymatic subunit (gp91^phox^), and the Rac G-protein [31]. After a pathogenic stimulus, the different subunits of the NADPH oxidase associate, leading to its activation and ROS production [32]. The expression levels of p22^phox^, p47^phox^, and gp91^phox^ subunits of the enzyme were measured by RT-PCR to study the effect of HT on the LPS- and α-syn-induced oxidative stress. mRNA levels of all subunits significantly increased after LPS and α-syn treatment with respect to control levels (Figure 4; *p* < 0.05). Treatment with HT prevented these increases in most cases (*p* < 0.05, Figure 4). The upcoming experiments were performed with the dose of 50 µM, as previously stated, because it was the highest non-toxic and most effective dose in most cases.

As NADPH oxidase is the main source of ROS in microglial cells, we measured the level of oxidative stress after LPS or α-syn treatment and the effect of HT on ROS generation. For this purpose, we took advantage of a fluorogenic probe that exhibits fluorescence signal in the presence of ROS. LPS was able to induce a three-fold increase of ROS production (2.92-fold change with respect to the control, *p* < 0.05; Figure 5A,B,D). However, this production was not significantly reduced by HT (Figure 5C,D). In the case of α-syn, a strong induction of ROS production was observed when cells were treated (4.02-fold change with respect to the control, *p* < 0.001; Figure 5E–H). This induction was significantly reduced when HT was added to the cells (2.09-fold change with respect to the control, *p* < 0.001; Figure 5F–H).

### 3.4. HT Effects on Activation of AKT and MAPKs

As a further step, activation of protein kinase B (AKT) and MAPKs c-Jun N-terminal kinases (JNK) 1 and 2 and p38 was determined by Western blot analysis using specific antibodies against their phosphorylated and non-phosphorylated forms. In all the treatments tested, no changes were observed in the non-phosphorylated forms of the different proteins (Figure 6A,C,E,G), but this was not the case for the phosphorylated ones. As expected, both LPS and α-syn induced the expression of p-AKT (2.09- and 2.25-fold change, respectively, compared with the control; *p* < 0.05; Figure 6B). HT significantly reduced the α-syn-induced levels of P-AKT (149.3% ± 89.8%, with respect to the control; *p* < 0.05; Figure 6B).

Regarding MAPKs, LPS, but not α-syn induced the expression of both phosphorylated forms of JNK (28.88-fold change for pJNK1 (*p* < 0.01) and 8.37-fold change for P-JNK2 (*p* < 0.05), with respect to the control; Figure 6D,F). Remarkably, HT abolished both the LPS- and α-syn-inducing effect of P-JNK1 (6.05-fold change, respectively, with respect to the control; *p* < 0.01; Figure 6D). Contrary to both JNKs, the phosphorylated form of p38 was induced by both LPS and α-syn (2.34- and 3.05-fold change, respectively, with respect to the control; *p* < 0.001; Figure 6H), whereas HT reduced the expression in both cases (1.41-fold change ± 21.82% for LPS and 1.80-fold change ± 42.01% for α-syn, with respect to the control; *p* < 0.001; Figure 6H). These results suggest that MAPKs and AKT pathways could be potential pathways for the anti-inflammatory effect of HT.

### 3.5. HT Effect on Translocation of NF-κB to the Nucleus

LPS is one of the many signals that induce inflammatory activation through TLR engagement and nuclear translocation of NFκB p65. We wanted to know if α-syn also stimulates this transduction pathway and if the reduction of microglial activation by HT seen above for both stimuli was because of its effect on this pathway. For this sake, immunofluorescence against NF-κB was carried out.

As can be observed in Figure 7, both LPS and α-syn induced the activation and translocation to the nucleus of NF-κB p65 (Figure 7A; *p* < 0.05). HT treatment prevented translocation in response to LPS, but not for α-syn (Figure 7B; *p* < 0.05). This suggests the α-syn triggers a different signal transduction pathway to activate NF-κB that HT is not able to interfere with, as in the case of LPS. 

### 3.6. HT Effect on the Inflammasome Pathway

The inflammasome pathway is one of the culprits of microglial activation, as it results in the release of the previously induced IL-1β. This activation begins with a first signal (“priming stage”) that promotes the expression of different components of the inflammasome molecular platform (i.e., NLRP3). A second signal, usually ATP, is needed for inflammasome assembly with the protein adaptor ASC, as well as pro-caspase-1. This results in autoprocessing and cleavage of pro-caspase-1, which in turn cleaves pro-IL1β into IL1β, which will be released. Thus, for testing if LPS, α-syn, and HT affected this signaling pathway, we next separately applied these agents as the first signal and ATP as the second signal, and sought to measure NLRP3 mRNA levels (by RT-PCR analysis) and IL-1β protein levels (by Western blot analysis).

Both LPS and α-syn increased the expression level of NLRP3 (6.64-fold change and 4.55-fold change, respectively, compared with the control; Figure 8A and B; *p* < 0.01 in both cases). HT treatment reduced significantly NLRP3 mRNA levels induced by LPS (3.53-fold change compared with the control; *p* < 0.01; Figure 8A), but not by α-syn. Regarding IL-1β protein levels, although both stimuli induced the expression of the NLRP3, only LPS produced an increase in IL-1β levels (3.71-fold change with respect to the control; *p* < 0.05; Figure 8C,D). HT reduced IL-1β protein levels induced by LPS, although contrary to NLRP3, did not reach significance (Figure 8C). Regarding the effect of α-syn on IL-1β protein levels, it seems that other factors may be needed to induce IL-1β protein maturation.

## 4. Discussion

PD is a high prevalent neurodegenerative disorder that affects 10 million people all over the world [33]. Although pharmacological interventions improve the quality of life of PD patients, this brain disorder has still no cure. In recent years, several non-pharmacological strategies aimed to reduce the incidence or slow down the disease progression have been under development. Among them, dietary interventions have gained much attention since the demonstration that PD incidence could be influenced by environmental factors, among them diet. In this sense, MD is the one that has shown better results [34]. A main component of MD is HT, a polyphenol present in virgin olive oil. Previous studies have shown the antioxidant and anti-inflammatory properties of HT in several models of human diseases, such as rheumatoid arthritis, hepatic and intestinal inflammation, diabetes, glioma, and osteoarthritis, among others [34]. In vitro studies have also revealed the anti-inflammatory effects of HT in monocytes/macrophages and point out some of its possible mechanism of action [17,18,19]. However, the possible anti-inflammatory properties of HT in microglia, the main immune cells in the CNS, have remained unexplored.

In this work, we provide compelling evidence that HT is able to reduce the microglial activation induced by two different insults: (i) LPS and (ii) α-syn fibrils, the pathological form of this protein. LPS, the active immunostimulant in the cell wall of Gram-negative bacteria, has been widely used as a potent pro-inflammatory stimulus for microglial cells, and its mechanism of action is well characterized [35]. On the other hand, α-syn is the main component of Lewy bodies, inclusions of proteins whose presence is closely associated with cell death of nigrostriatal dopaminergic neurons characteristic of PD [1]. With this dual approach, we sought to characterize the potential anti-inflammatory effects of HT in BV2 cells.

To evaluate the anti-inflammatory effects of HT, we used PCR to determine if HT was able to decrease the expression levels of several pro-inflammatory mediators induced by both LPS and α-syn stimulated conditions. Our results show that HT treatment decreases the mRNA levels of all the pro-inflammatory mediators studied, including TNF-α, IL-1β, iNOS, IL-6, and CXCL10. These results are of high interest as the levels of cytokines such as TNF-α, IL-1β, and IL-6 are increased in Parkinsonian brains and serum [35,36]. It has been evidenced the role of TNF-α as a key neuroinflammatory mediator of neurotoxicity and neurodegeneration in animal models of PD [37]. In addition, a potential deleterious role for NO in PD has also been demonstrated. Hence, in the 1-methyl-4-phenyl-1,2,3,6-tetrahydropyridine (MPTP) rodent model of PD, dopaminergic neurons were protected by the treatment with non-selective NOS inhibitors or gene deletion of the iNOS gene [38]. CXCL10 has also been implicated in numerous neuropathologies, including those involved in abnormal α-syn deposition, such as PD, multiple sclerosis, and human immunodeficiency virus (HIV) dementia [39].

These results point out the ability of HT to downregulate the activation state of microglial cells. Different in vivo and in vitro studies have highlighted that the inhibition of inflammatory response of activated microglia is a promising approach to treat various neurodegenerative disorders like PD, where neuroinflammation plays a crucial role [40]. The decrease in the pro-inflammatory mediators induced by HT in our model suggests that diets containing this polyphenol or its precursors could decrease the inflammatory environment of the brain, leading to a possible decrease in PD incidence and/or progression.

Once we demonstrated the anti-inflammatory effect of HT in LPS and α-syn BV2-treated cells, we wanted to elucidate the molecular mechanism responsible of such an effect. Various pathways, including NF-κB, MAPK, and inflammasomes, are known to be involved in the transcriptional regulation of inflammatory mediators [41]. Thus, it is known that, during the inflammatory response, extracellular LPS is recognized by TLR4, one of the pattern recognition receptors that activates multiple signal transduction pathways such as those related to MAPKs and NF-κB. The innate immune response is also triggered by intracellular inflammasomes, whose activation produces cleavage and release of pro-inflammatory cytokines such as IL-1β and IL-18. On the other hand, abnormal α-syn protein aggregates can activate microglial cells. As stated above, this sustained microglia activation contributes to the pathogenic processes of PD. It is known that TLR4 also recognizes α-syn and activates downstream signaling mechanisms, leading to the release of pro-inflammatory mediators that are counterbalanced by Nurr1 expression [42].

Keeping this information in mind, we wanted to know which of the mentioned pathways were affected by HT after LPS and α-syn treatment. For this purpose, we performed Western blot to study the levels of several MAPKs, such as JNK 1/2 and p38, and the levels of AKT. Our results show that the treatment with both LPS and α-syn induced an increase in the phosphorylated forms of JNK 1/2, p38, and AKT that was further decreased by HT treatment. MAPKs pathway is a critical axis, essential for both induction and propagation of the inflammatory response after LPS challenge in macrophages. The increased activity of MAPKs and AKT as well as their involvement in the regulation of the inflammation mediator synthesis at the transcription level and translation makes them potential targets for anti-inflammatory therapy [43].

A typical feature of TLR ligation is activation of NF-κB, leading to transcription of pro-inflammatory genes [38]. LPS is known to also induce I-κB phosphorylation, through the activation of kinases such as MAPK p38 and JNK; all of them are involved in activating key transcription factors, including NFκB [44]. Activation of NF-κB relies on activation of the IKK complex, in which inhibitor of nuclear factor kappa-B kinase subunit β (IKK-β) triggers the canonical pathway. We have previously shown that LPS induces IKK-β expression in reactive microglia in a similar way that iNOS does, as studied by Western blot and qPCR [28]. In this study, we wanted to know if HT was able to reduce nuclear translocation of p65 NFκB after stimulating BV2 microglial cells with either LPS or α-syn. Our results show that NF-κB is translocated to the nucleus after LPS and α-syn insults. Interestingly, HT treatment was able to decrease this effect in LPS-treated BV2 cells. However, this effect was not seen when cells where treated with α-syn.

Another important mechanism of action of neurodegeneration as part of the neuroinflammation process is the induction of oxidative stress. In microglial cells, the main source of ROS is the NADPH oxidase enzyme. We thus wanted to know if HT was able to counteract the induction of this enzyme and the consequent generation of ROS in our two models of microglial activation. Interestingly, our results showed a reduction in the expression levels of the mRNA of the different subunits of the NADPH oxidase and a decrease in the ROS production when HT was added to the LPS and α-syn-activated microglia, although not significantly for LPS. These finding are relevant because oxidative stress is one of the most important features in main neurodegenerative diseases, including PD. It is also known that the dopaminergic neurons of the substantia nigra are a population of cells especially vulnerable to oxidative stress owing to the oxidative metabolism of dopamine and the presence of iron that can generate ROS through the Haber–Weiss and Fenton reactions. Therefore, any strategy capable of decreasing oxidative stress in the brain, as could be the intake of HT in the diet, could protect the brain, and hence reduce the incidence of neurodegenerative diseases.

Finally, another route that could be involved in the anti-inflammatory properties exerted by HT is the one controlled by the NLRP3 inflammasome. This molecular platform mediates inflammatory responses through maturation processing and the release of IL-1β in a Caspase 1-dependent mechanism [30]. NLRP3 is one of the most intensively investigated inflammasomes, controlling disease progression and inflammatory responses [45]. It is known that LPS (priming response) followed by ATP treatment (activation response) is able to activate the NLRP3 inflammasome, inducing the release of IL-1β [30,45]. Activation of the NLRP3 inflammasome requires the activation of TLR4/NFκB signaling pathways [42]. We thus measured both the expression levels of the NLRP3 using qPCR (priming response) and the production of mature IL-1β (activation response). Our data demonstrate that LPS, but not α-syn was able to induce the assembly of this molecular platform, leading to the production of IL-1β. This activation was, however, decreased when BV2 cells were co-treated with HT.

## 5. Conclusions

In summary, HT was able to prevent, totally or partially, most of the immune-associated alterations induced by LPS. Thus, microglial activation, expression of NADPH oxidase and MAPKs, production of ROS, and activation of the inflammasome induced by LPS were reduced or prevented by HT. HT was also able to decrease the activation of microglial cells after α-syn treatment. This effect seems to be mediated by MAPKs and the generation of ROS through the NADPH oxidase. The different results found for LPS and α-syn suggest pleiotropic anti-inflammatory effects of HT. Some of these differences could be because of the ability of HT to prevent α-syn aggregation, and thus to counteract α-syn-induced toxicity in PC12 cell cultures [46,47].

The results discussed so far support the use of HT to prevent the inflammation associated with PD and shed light on the relationship between MD, rich in HT and other polyphenols, and PD.

## Figures and Tables

**Figure 1 antioxidants-09-00036-f001:**
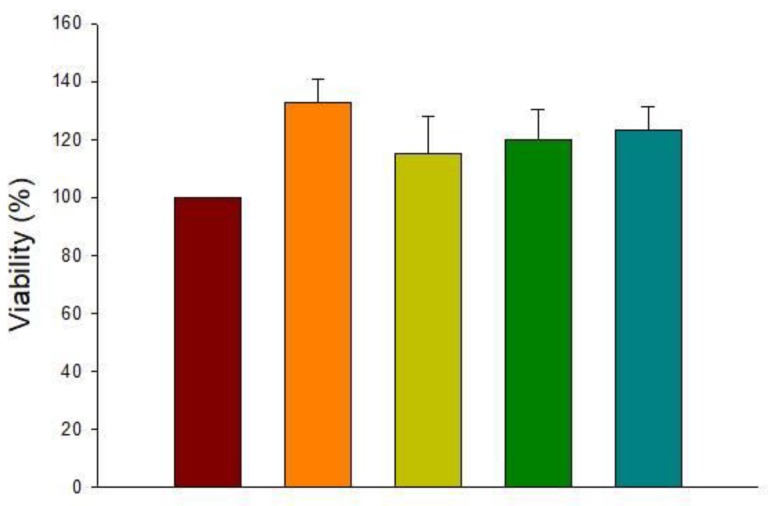
Effect of hydroxytyrosol (HT) (1, 10, 25, and 50 μM) on BV2 cell viability (%). Data are expressed as mean ± SEM (*n* = 3).

**Figure 2 antioxidants-09-00036-f002:**
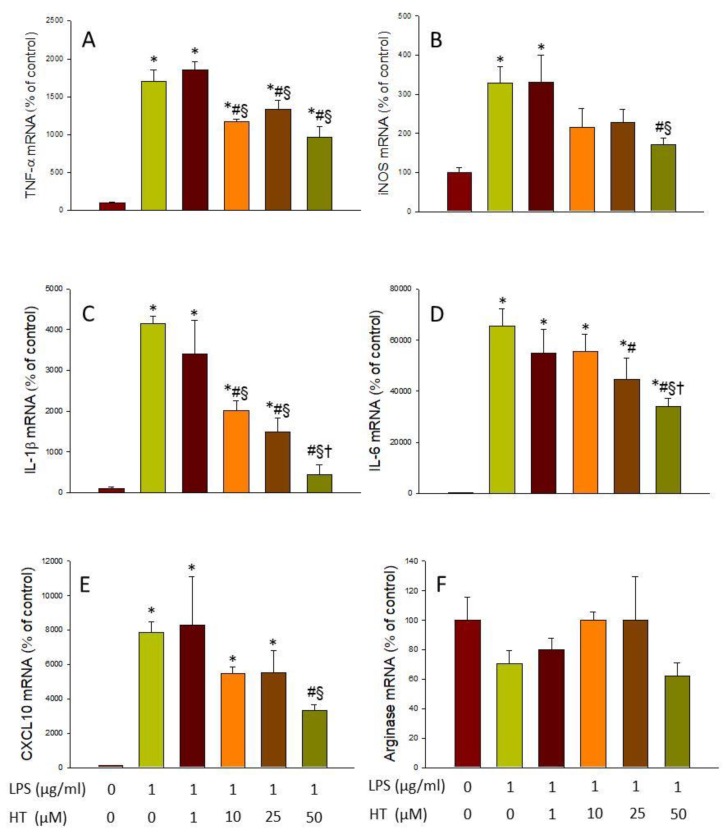
Real-time RT-PCR. (**A**) tumor necrosis factor (TNF)-α gene expression; (**B**) inducible nitric oxide synthase (iNOS) gene expression; (**C**) interleukin (IL)-1β gene expression; (**D**) IL-6 gene expression; (**E**) CXCL10 gene expression; (**F**) Arginase gene expression, after the treatment with lipopolysaccharide (LPS) alone (1 mg/mL) or with HT (1, 10, 25, and 50 μM). Data are expressed as mean ± SEM (*n* = 3), normalized to β-actin, and expressed as percentage relative to the control group. Statistical significance (one-way analysis of variance (ANOVA) followed by the LSD post hoc test for multiple comparisons): * compared with the control group; # compared with the LPS group; § compared with the LPS + HT 1 μM group; † compared with the LPS + HT 10 μM group; *p* < 0.05.

**Figure 3 antioxidants-09-00036-f003:**
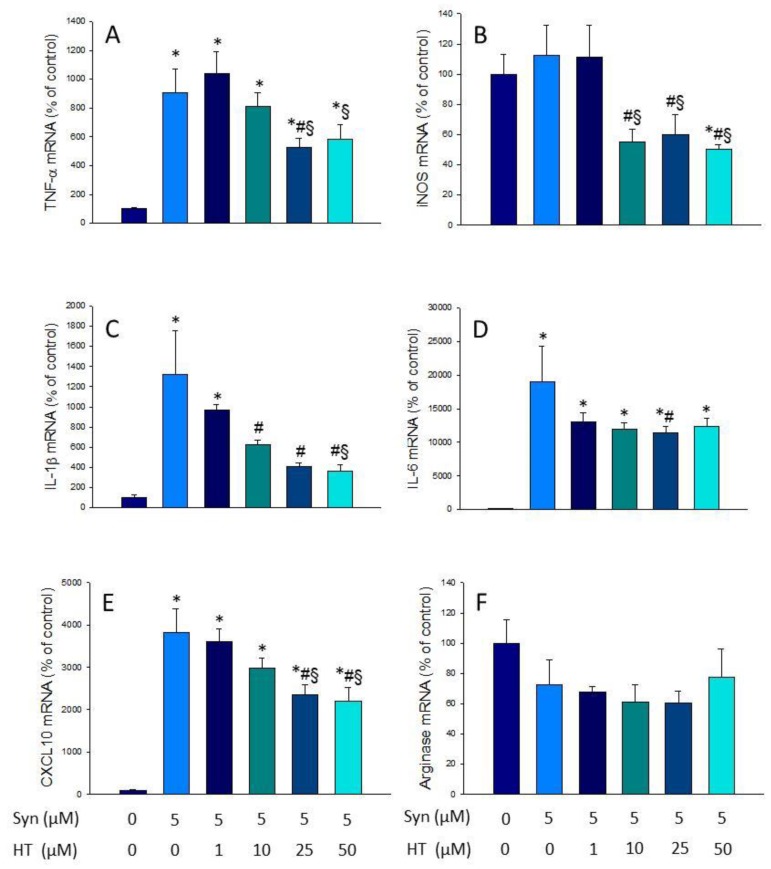
Real-time RT-PCR. (**A**) TNF-α gene expression; (**B**) iNOS gene expression; (**C**) IL-1β gene expression; (**D**) IL-6 gene expression; (**E**) CXCL10 gene expression; (**F**) Arginase gene expression, after the treatment with α-syn alone (5 μM) or with HT (1, 10, 25, and 50 μM). Data are expressed as mean ± SEM (*n* = 3), normalized to β-actin, and expressed as percentage relative to the control group. Statistical significance (one-way ANOVA followed by the LSD post hoc test for multiple comparisons): * compared with the control group; # compared with the Syn group; § compared with the Syn + HT 1 μM group; *p* < 0.01.

**Figure 4 antioxidants-09-00036-f004:**
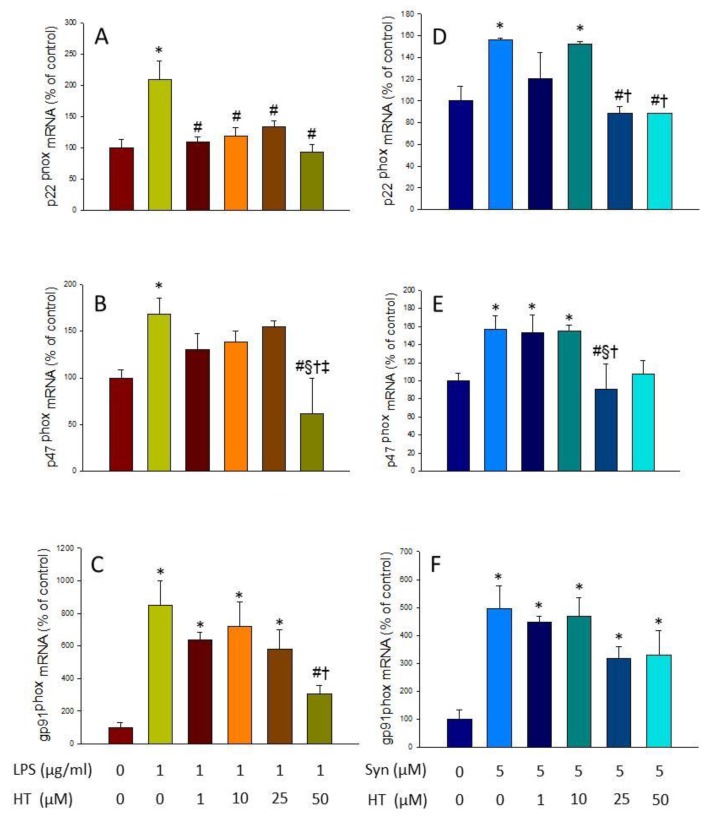
Real-time RT-PCR. (**A**) p22phox gene expression; (**B**) p47phox gene expression; (**C**) gp91phox gene expression, after the treatment with LPS alone (1 mg/mL) or with HT (1, 10, 25, and 50 μM); (**D**) p22phox gene expression; (**E**) p47phox gene expression; (**F**) gp91phox gene expression, after the treatment with α-Syn incubated alone (5 μM) or with HT (1, 10, 25, and 50 μM). Data are expressed as mean ± SEM (*n* = 3), normalized to β-actin, and expressed as percentage relative to the control group. Statistical significance (one-way ANOVA followed by the LSD post hoc test for multiple comparisons): * compared with the control group; # compared with the LPS/syn group; § compared with the LPS/syn + HT 1 μM group; † compared with the LPS/syn + HT 10 μM group; ‡ compared with the LPS + HT 25 μM group; *p* < 0.05.

**Figure 5 antioxidants-09-00036-f005:**
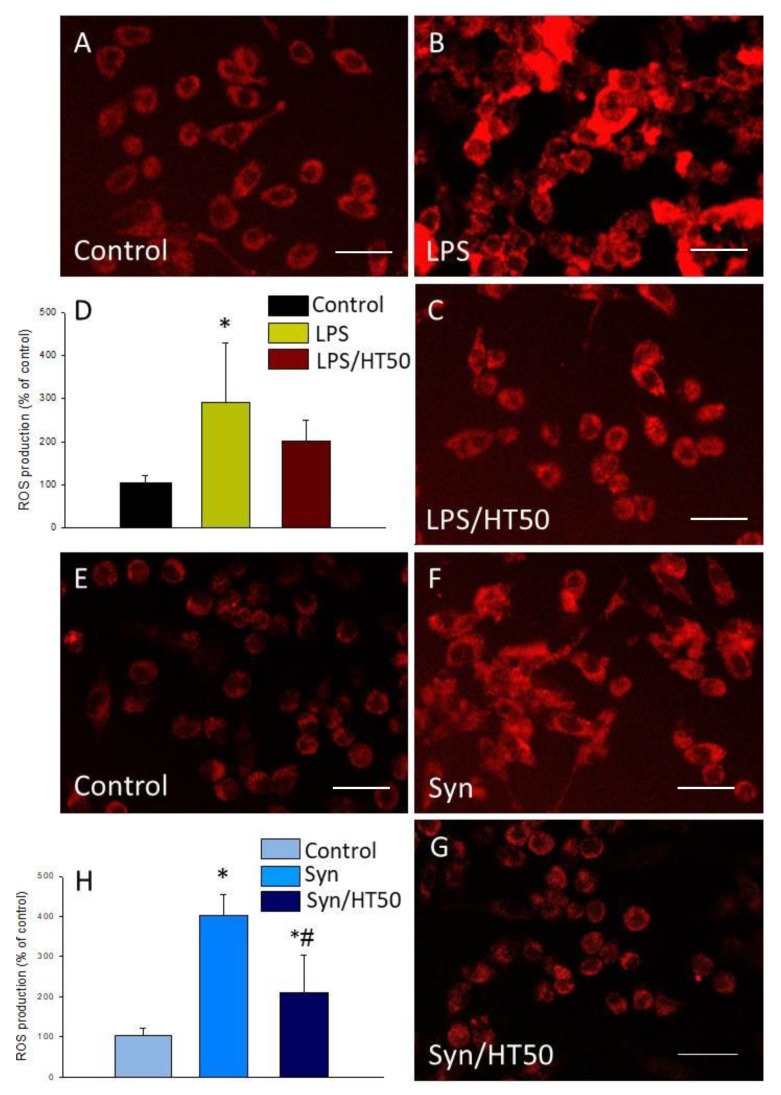
Reactive oxygen species (ROS) production. ROS were measured 6 h after the different treatments assayed. (**A**) Control (BV2 cells); (**B**) LPS (1 mg/mL); (**C**) LPS (1 mg/mL) plus HT (50 μM). (**D**) Quantification of ROS production with LPS with or without HT. Results are mean ± SEM (*n* = 4), expressed as percentage of control. Scale bar, 50 µm. (**E**) Control (BV2 cells); (**F**) α-syn (5 μM); (**G**) α-syn (5 μM) plus HT (50 μM); (**H**) quantification of ROS production with α-syn, with or without HT. Results are mean ± SEM (*n* = 4). Results are expressed as percentage relative to the control group. Statistical significance (one-way ANOVA followed by the LSD post hoc test for multiple comparisons): * compared with the control group; # compared with the syn group; *p* < 0.05 (for LPS), *p* < 0.001 (for α-syn). Scale bar, 50 µm.

**Figure 6 antioxidants-09-00036-f006:**
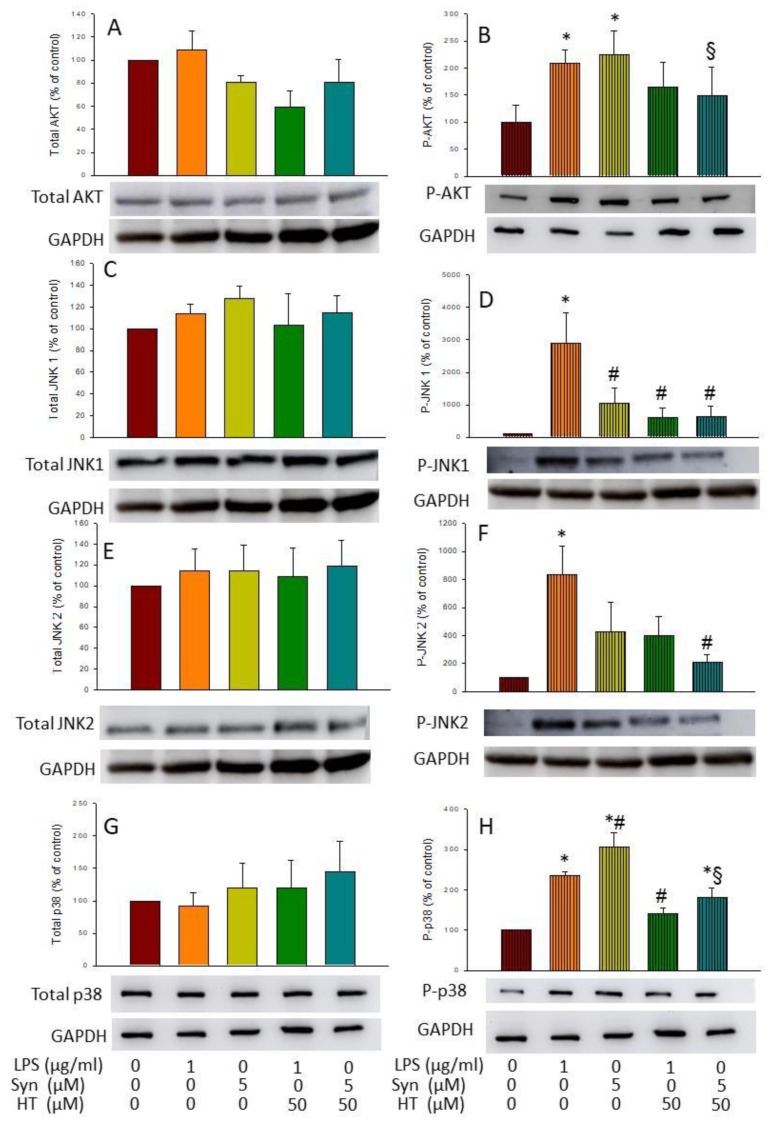
Blot. (**A**) Total AKT; (**B**) P-AKT; (**C**) total JNK 1; (**D**) P-JNK 1; (**E**) total JNK 2; (**F**) P-JNK 2; (**G**) total p38; (**H**) P-p38 after 6 h of treatment (control (BV2 cells), LPS (1 mg/mL), α-syn (5 μM), LPS (1 mg/mL) + HT (50 μM), and α-syn (5 μM) + HT (50 μM). GAPDH is used as housekeeping. Data are expressed as mean ± SEM (*n* = 3), normalized to GAPDH expression, and relative to the control group. Statistical significance (one-way ANOVA followed by the LSD post hoc test for multiple comparisons): * compared with the control group; # compared with the LPS group; § compared with the syn group; *p* < 0.05.

**Figure 7 antioxidants-09-00036-f007:**
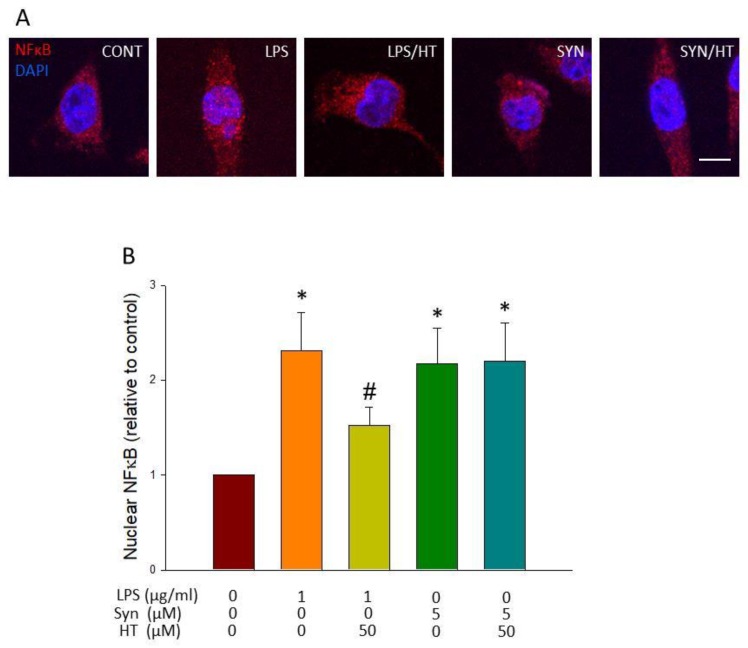
Analysis of nuclear translocation of NF-κB p65. (**A**) Immunostaining of NF-κB p65 of control and treated cells showing its distribution in cytosolic and nuclear compartments. (**B**) Quantitative assessment of NF-κB p65 nuclear translocation is shown as fold change of nuclear staining in treated versus control cells (*n* = 4). Results are mean ± SEM, relative to control group. Statistical significance (one-way ANOVA followed by the LSD post hoc test for multiple comparisons): * compared with the control group; # compared with the LPS group; *p* < 0.05. Scale bar, 5 μm.

**Figure 8 antioxidants-09-00036-f008:**
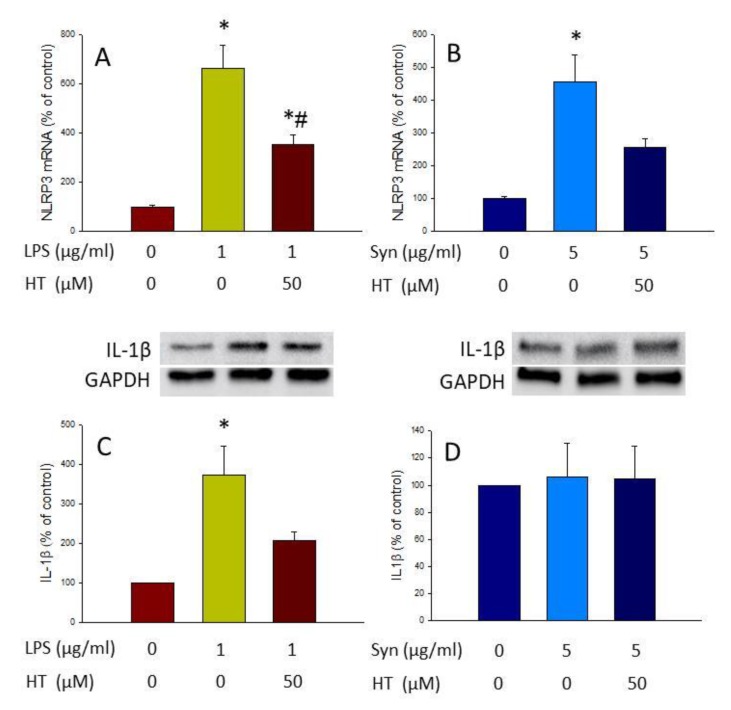
Inflammasome activation was measured by RT-PCR of NLRP3 and Western blot of IL-1β NLRP3 gene expression after treatment with LPS and LPS plus HT (**A**), α-syn and α-syn plus HT (**B**), IL-1β production after treatment with LPS and LPS plus HT (**C**), and α-syn and α-syn plus HT (**D**). For Western blot, cells were treated with LPS or α-syn fibrils, with or without HT (50 µM), for 3 h, and then with ATP (1 mM, Sigma-Aldrich, San Luis, MO, USA) for 1 h as the second signal for inflammasome activation. Data are expressed as mean ± SEM (*n* = 3). For PCR assay, the results are normalized to β-actin and expressed as percentage relative to the control group. Statistical significance (one-way ANOVA followed by the LSD post hoc test for multiple comparisons): * compared with the control group; # compared with the LPS group; *p* < 0.01. For Western blot assay, data are normalized to GAPDH expression and relative to the control group. Statistical significance (one-way ANOVA followed by the LSD post hoc test for multiple comparisons): * compared with the control group; *p* < 0.01.

**Table 1 antioxidants-09-00036-t001:** RT-PCR primers. IL, interleukin; TNF, tumor necrosis factor; iNOS, inducible nitric oxide synthase.

mRNA	Primers
IL-1β	F: 5′-TTGACGGACCCCAAAAGATG-3′ R: 5′-AGAAGGTGCTCATGTCCTCA-3′
TNF-α	F: 5′-AGCCCACGTCGTAGCAAACCACCAA-3′ R: 5′-AACACCCATTCCCTTCACAGAGCAAT-3′
iNOS	F: 5′-CTTTGCCACGGACGAGAC-3′ R: 5′-TCATTGTACTCTGAGGGCTGAC-3′
Arginase	F: 5′-TCACCTGAGCTTTGATGTCG -3′ R: 5′-CTGAAAGGAGCCCTGTCTTG -3′
β-actin	F: 5′-CCACACCCGCCACCAGTTCG-3′ R: 5′-CCCATTCCCACCATCACACC-3′
CXCL10	F: 5′-AAGCATGTGGAGGTGCGAC-3′ R: 5′-CTAGGGAGGACAAGGAGGGT-3′
IL-6	F: 5′-GACAAAGCCAGAGTCCTTCAGA-3′ R: 5′-AGGAGAGCAATTGGAAATTGGGG-3′
NLRP3	F: ACCAGCCAGAGTGGAATGAC-3′ R: ATGGAGATGCGGGAGAGATA-3′
p22^phox^	F: 5′-GAATTCCGATGGGCAGATCGA-3′ R: 5′-GGAQTCCCGTCACACGACCTCA-3′
P47^phox^	F: 5′-ATTTGGAGCCCTTGACAG-3′ R: 5′-GATGGTTACATACGGTTCACCTG-3′
gp91^phox^	F:5′-GCACAGCCAGTAGAAGTAGATCTTT-3′ R: 5′-GCTGGGATTGGAGTCACG-3′

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
