# Peer review of "Hydroxytyrosol Decreases LPS- and α-Synuclein-Induced Microglial Activation In Vitro"

_antioxidants, 2019, doi:10.3390/antiox9010036_

Round 1

Reviewer 1 Report

In this paper "Hydroxytyrosol decreases LPS- and 3 α-synuclein-induced microglial activation in vitro" Gallardo-Fernández et al describe a series of experiments examining the possible molecular mechanisms involved in the anti-inflammatory effect of Hydroxytyrosol under stress conditions using both LPS and aggregated α-syn as stressors.

Introduction.
The introduction is well written and appropriate. There are a number of typos/spacing errors that should be fixed for example line 64 "(iNOS)[7].However," no space after stop. Please check and correct.

Materials and methods.
In general the methods are well describe however additional info would be helpful. Also check spelling, grammar and formatting e.g. line 134 "RNA using Revert Aid First Strand cDNA Synthesis Kit". Line 150 "seeding concentration of 150 ×103 cell/well" Line 165 "incubated for 1hat room temperature"

1. What was the range of passage number for cell cultures
2. Provide rational for choose time course of treatments e.g. line 108 "without HT (1, 10, 25 and 50 μM, Sigma Aldrich, Steinheim, Germany) for 6h."
3. How were stocks of LPS, α-syn and HT prepared, what was the solvent ? And was solvent used in control treatments at appropriate levels.

Results.
Figure 1 appears to be insert into the text cutting off sentence 208 from 209
All figures could be improve by more attention to presentation.
For example
X axis change C, HT1 etc to 0, 1, 10 etc. Label axis as HT concentration (uM)
Y axis change label to Viability (%)
In the Legend needs to read something like "Effect of HT of BV2 cell viability. ........."

Figure 7 in particular is not well presented. The upper half of the figure presents panels of images but what is the purpose of the DAPI panel? It would be better to overlay the red image on the DAPI to more clear show translocation is present. The Lower half shows a bar chart labelled K. The figure legend is not very helpful also was tis the effect go HT alone on translocation ??

Discussion
The discussion is well written (check for spacing, spelling and grammar) but could be improved by spending less time focusing on a re-run of the result and devote space to the content of these results in the wider research effort. As a simple example how much HT is circulating in the average person on a traditional western diet versus the MD diet. Is the 50uM used in the later studies representative of high HT plasma concentration or no. is the incidence of PD reduced in Mediterranean countries ?

Reviewer 2 Report

This is an interesting study showing the beneficial effects of hydorxytyrosol on LPS- and α-synuclein-induced microglial inflammation in vitro. The authors investigated the mechanisms involved in these anti-inflammatory effects. The study is technically sound, the main conclusions are supported by data but some issues should be addressed by the authors.

My main concern is that the authors seem to obviate that the main circulating hydroxytyrosol metabolite is hydrotyrosol sulfate and this is invoved in the neuroptrotective properties of HT intake Molecules. 2019 May; 24(10): 2001. This point should be clearly addressed in the manuscript since they use HT (this is an important limitation of the study).

The authors should also justify that they are working at relevant physiological concentrations.

The use murine microglial BV2 cell line should be clearly justified. Why do they use this cell line?

Reviewer 3 Report

The current study aimed to determine the effects on hydroxytyrosol (HT), a key component of the Mediterranean Diet, on the neuro-inflammatory pathway/s as it relates to neurodegenerative diseases, such as Parkinson's Disease (PD). The results indicate that HT can significantly reduce various inflammatory markers related to PD and systematically delve into the mechanism thereof. Overall, the methods appear sound and the results are positive and well described. Nevertheless, I have various comments that I feel need to be addressed before the manuscript can be accepted for publication.

Overall, the English needs to be improved. I would recommend that the authors have a native English speaker proofread the manuscript. Section 2.1: The second paragraph of this section is a bit repetitive as many assays utilized similar treatments or analysis techniques. I think this section in particular, as it essentially describes the main experimental design, would benefit from more concise and clear writing. Section 3.1: This section seems to be cut off as the first paragraph stops mid-sentence. The authors should correct this error and provide the full result information. Furthermore, the authors should describe in the methods that viability was the primary metric that was used to test cell cytotoxicity and provide an explanation as to why all viability results were greater than 100%. Is there a particular reason as to why HT treatment would could such cell proliferation? Section 3.2: The authors state that "treatment with HT was able to reduce the expression of all [pro-inflammatory markers] in a dose-dependent manner  (p < 0.05; vs. LPS levels; Figures 2A-E)". First of all, I do not think stating "dose-dependent" manner is fair in this context as there were generally only significant effects between a few different doses and not all. I think it would be more fair to state that the ANOVA analysis indicated a significant effect of dose, if that is the case (which I assume it is since post-hoc testing was performed). Furthermore, from Fig. 2 I dont see a does-dependent reduction in expression from the LPS level when treated with HT. On the contrary, some of the LPS/HT1 values are actually larger than the LPS values (although not significantly). A more accurate statement would be something along the lines of "higher doses of HT" or "HT doses above X"..."significantly reduced expression compared to LPS alone". Section 3.3: My comments from section 3.2 directly apply here as well. In general, letters should not be used to indicate significance in the figure plots (bar graphs). These can make the figures confusing as the letters can interfere with the panel labels. I would recommend using more conventional indicators such as *, #, or multiple instances of these symbols. For all figures, the font size of axes labels and legends should be increased.

Round 2

Reviewer 2 Report

Accept in present form